# Functional, Structural, and AI-Based MRI Analysis: A Comprehensive Review of Recent Advances

**DOI:** 10.3390/diagnostics15243212

**Published:** 2025-12-16

**Authors:** Xingfeng Li

**Affiliations:** Department of Surgery and Cancer, Imperial College London, London W12 0HS, UK; xingfengli@gmail.com

**Keywords:** MR image analysis, fMRI, quantitative MRI, radiomics, machine/deep learning, segmentation, model selection

## Abstract

**Background/Objectives:** Since the invention of MRI, analytical methods for MRI data have continuously evolved. In recent years, the rapid development of artificial intelligence has transformed MRI data analysis—from functional MRI (fMRI) techniques to deep learning-based image segmentation, and from traditional machine learning to radiomics for clinical applications. **Methods:** This review provides a succinct summary of recent progress in fMRI and structural MRI analysis. The discussed techniques include fMRI, quantitative MRI (qMRI) methods such as T1 and T2 relaxation time mapping, and proton density imaging. Approaches for diffusion, perfusion, and the Dixon method are also described. Furthermore, studies published between 2012 and 2025 on MRI radiomics were reviewed. Different neural network architectures related to radiomics-based segmentation are compared and discussed. **Results:** A major trend in both fMRI and MRI analysis is the increasing use of quantitative methods, which enable better cross-study comparison and reproducibility. Deep learning remains to progress rapidly in MRI research, particularly in segmentation tasks, with new loss functions and network architectures developed to improve performance. These methods are expected to undergo further optimization and find broader applications in clinical practice. **Conclusions:** Despite substantial progress, challenges remain in standardization, validation, and clinical translation. Continued efforts are necessary before these advanced analytical techniques can be fully integrated into routine medical practice.

## 1. Introduction

In recent years, substantial progress has been made in the analysis of functional and structural magnetic resonance (MR) images, driven primarily by advances in computational modeling, machine learning, and large-scale data sharing. Functional MRI (fMRI) analysis has evolved beyond conventional activation mapping to more sophisticated network-level approaches, including functional connectivity, graph-theoretical analysis, and time-resolved dynamic modeling. Emerging research in brain and mind decoding based on fMRI has further demonstrated its potential for identifying cognitive states and mental processes.

Structural MRI has also advanced significantly through the development of improved segmentation algorithms, voxel- and surface-based morphometry, and diffusion imaging techniques that enable detailed mapping of white matter pathways. In addition, quantitative MRI (qMRI) and radiomics have been increasingly applied in clinical studies to detect subtle tissue alterations and derive imaging biomarkers.

MRI radiomics extracts quantitative features—such as texture, shape, and intensity patterns—that may reflect underlying tissue biology, including tumor heterogeneity, beyond what is visible to the human eye. Combined with machine learning or statistical modeling, radiomics supports diagnosis and prognosis, predicts treatment response, and contributes to precision medicine by transforming images into mineable data for clinical decision-making.

Collectively, these methodological developments have enhanced the precision, reproducibility, and scalability of MRI-based research, facilitating the identification of disease-specific imaging markers and the integration of multimodal data. This review summarizes recent computational approaches for analyzing functional and structural MRI data, highlighting their applications in neuroscience and clinical research. It also discusses key challenges that must be addressed to achieve robust and clinically translatable outcomes. The aim is to provide an overview of current MRI data analysis methods and to emphasize emerging trends shaping the future of this field.

## 2. fMRI Data Analysis

This review was conducted through a comprehensive literature search on PubMed and Google Scholar in September 2025. The search terms included “fMRI”, “MRI radiomics”, “MR image segmentation”, and “quantitative MRI.” Relevant articles were identified and assessed, and data were extracted from selected studies for inclusion in this review. This review covers fMRI, quantitative MRI, image segmentation, and radiomics studies.

Blood Oxygenation Level-Dependent (BOLD) contrast is the most widely used mechanism in fMRI, allowing researchers to noninvasively infer brain activity [1]. To analyze fMRI signals, the first step after data collection is pre-processing of the fMRI time series. This often includes motion correction to remove artifacts caused by subject movement during scanning [2], as well as correction for physiological confounds such as low-frequency respiratory and cardiac fluctuations and respiratory motion [3]. More recently, deep learning methods have also been applied to fMRI pre-processing [4].

Following pre-processing, analysis methods are selected according to the study objective. For activation detection, hierarchical approaches employing mixed-effects models have been widely adopted. In these models, first-level (single-run) and second-level (combining of several runs or group level comparison) analyses are integrated into a single framework for activation detection [5].

For connectivity studies, both functional and dynamic functional connectivity methods—based on sliding time windows—have been applied in clinical research [6]. These approaches are implemented for both resting-state and task-based fMRI. Other network-level methods, such as independent component analysis (ICA) and principal component analysis (PCA), have also been employed. To examine effective connectivity, techniques including Granger causality [7], nonlinear system identification [8], and multiple moving averaging methods [9] have been realized to investigate information flow within networks.

Currently, BOLD contrast is most commonly measured using single-shot gradient-echo-planar imaging (EPI). Recently, multi-echo (ME) fMRI has been proposed to improve signal fidelity by differentiating the origins of fMRI signals [10]. Comparisons between ME and optimized single-echo EPI for task-based fMRI have shown that ME-fMRI does not always outperform single-echo schemes [11]. While single-echo fMRI offers higher statistical power, ME-fMRI demonstrates superior reliability at the single-subject level and has been applied in connectivity studies [12,13]. Tensor ICA can be used to analyze ME-fMRI by decomposing data across time, space, and echo times (TEs), which allows separation of BOLD and non-BOLD components [11,14].

Beyond activation and connectivity studies, decoding mental and perceptual states using fMRI has gained popularity over the past two decades, with numerous high-impact publications [15]. Brain decoding investigates the potential involvement of cognitive functions based on observed brain activity. Methods include spatial comparisons to meta-analytic maps, with applications ranging from letter and number recognition [16] to glioblastoma heterogeneity [17] and Parkinson’s disease [18]. Network-based approaches, resting-state decoding, machine learning methods, and deep learning autoencoder models have also been employed [19,20,21]. For comprehensive reviews on fMRI-based decoding, see [20].

Despite its promise, brain decoding faces logical and ethical limitations. Critics argue that decoding models cannot fully disentangle neural mechanisms from their epiphenomena and that constructing decoders consumes resources that might be better spent on other scientific pursuits [15,22]. Moreover, inferring mental properties from physiological or biometric signals raises legal and ethical concerns. There are calls to establish norms prohibiting non-consensual biometric mind-reading, ensuring that certain aspects of mental life remain protected from intrusive probing without consent [22].

## 3. Quantitative MRI (qMRI)

Weighted MR images are primarily qualitative and lack direct physical meaning, whereas quantitative MRI (qMRI) generates numerical maps derived from signal changes over time [23]. Weighted MRI produces images with varying contrast by emphasizing specific tissue properties for anatomical evaluation, such as T1-weighted or T2-weighted scans. In contrast, qMRI provides objective, physically meaningful measurements, including T1 and T2 relaxation times. Recent advances in qMRI have enabled more accurate and reproducible assessment of tissue properties, overcoming the limitations of weighted MRI, which is dependent on scanner settings. By mapping tissue characteristics in physical units, qMRI offers precise insights into microstructural changes, enhancing diagnostic accuracy and improving understanding of disease mechanisms [24]. Commonly used qMRI measures include T1, T2, T2*, and proton density, which are increasingly applied for disease detection, monitoring, and research.

Several methods exist for estimating T1 longitudinal relaxation times [25,26]. One commonly used approach is the variable flip angle method, in which two acquisitions with different nominal excitation angles (e.g., 3° and 20°) are performed [27,28,29]. In addition, the transmitted radiofrequency field is measured for B1 mapping. In this method, two flip-angle images (e.g., α_1_ = 60° and α_2_ = 120°) are acquired to estimate the B1 map [30]. T2*/T2 imaging is typically obtained using a multi-echo (ME) sequence, in which images at different TEs are collected to estimate T2*/T2 [31,32]. Based on these measurements, T1 and proton density (PD) maps can be calculated [27]. Figure 1a–d shows the images required for T1 calculation, while Figure 1e,f display an example of the resulting T1 and PD maps, respectively.

T1 relaxation times are sensitive to changes in tissue water content or local molecular environment, such as in edema, making them valuable for quantifying pathological changes. T1 mapping can also detect and measure diffuse fibrosis at early stages [33,34,35]. T2* measurements enable early disease detection, such as quantifying total body iron in patients with iron overload [36,37].

As an example, for brain tumour study, it showed that qMRI has potential to improve disease characterization in brain tumor patients under tight clinical time-constraints [38,39,40]. The study demonstrated that T1 mapping could be used clinically for glioma grading, given its high performance. With logistic regression, another study showed that T2 mapping also improve for glioma grading [40].

For water, fat, and bone marrow studies, Dixon techniques are commonly employed [41,42]. By acquiring in-phase and out-of-phase images, water-only and fat-only images can be reconstructed, allowing the estimation of proton density fat fraction as a non-invasive quantitative biomarker [43,44,45]. The original 2-point Dixon method used only two images (in-phase and out-of-phase), but it was sensitive to magnetic field inhomogeneity [46]. To remedy this, the 3-point (multiple-point) method adds a third echo time, allowing the system to estimate and correct for field inhomogeneities, greatly improving robustness [47]. Multipoint Dixon sequences, such as six-point acquisitions, allow precise assessment of fatty muscle degeneration, with potential implications for treatment planning [48].

MRI perfusion techniques are widely used to diagnose and monitor neurological conditions, including brain tumors, stroke, dementia, and traumatic brain injury [49]. Common approaches include dynamic susceptibility contrast (DSC) MRI, dynamic contrast-enhanced (DCE) MRI, and arterial spin labeling (ASL). DSC-MRI, which tracks a gadolinium-based contrast agent, is the most widely available method, providing hemodynamic metrics such as cerebral blood flow, blood volume, mean transit time, and Tmax [50]. DCE-MRI similarly acquires rapid successive images after contrast injection, allowing analysis of “wash-in” and “wash-out” kinetics to improve tumor and vascular lesion delineation [51,52,53]. Radiomic features from DCE-MRI have been applied for classification studies [54]. ASL, in contrast, is a non-contrast technique that can generate quantitative perfusion maps and has been applied for disease diagnosis and monitoring, including comparative studies with DSC-MRI in glioma [55,56]. For additional perfusion MRI studies, see [57].

Diffusion imaging plays a critical role in structural MRI analysis, particularly for exploring tissue microstructure in the brain. Diffusion-weighted imaging (DWI) enables estimation of quantitative metrics such as the apparent diffusion coefficient. Diffusion tensor imaging (DTI) further models directional water diffusion along fiber tracts, allowing inference of fiber orientation. At least six DWI acquisitions with different orientations are required to estimate the diffusion tensor. [58]. Eigenvalues and eigenvectors derived from the tensor enable calculation of fractional anisotropy and mean diffusivity maps [59]. Conventional DTI cannot resolve crossing fibers; thus, high angular resolution diffusion imaging with spherical harmonics fitting has been developed to overcome this limitation [60]. DTI acquisitions can use single- or multi-shell b-values, with multi-shell commonly used in advanced diffusion studies and diffusion spectrum imaging [61,62].

## 4. MRI Radiomics Study

One approach to MRI data analysis is radiomics, which involves the high-throughput extraction of quantitative features from MR images that capture tissue shape, intensity, texture, and higher-order patterns [63]. These features can be used to build predictive or prognostic models for diagnosis, treatment response, or outcome prediction. Several surveys have summarized the field [64].

In recent years, there has been a marked increase in publications on MRI radiomics. A PubMed search using the keywords ‘MRI’ and ‘radiomics’ identified 6596 records as of 7 December 2025. After excluding animal studies using the query (MRI) AND (radiomics) NOT (animals), 6524 publications remained, indicating that 72 involved animal research and were therefore excluded from this review. Figure 2 presents the annual distribution of publications retrieved from PubMed. As shown, the number of studies continues to rise, reflecting sustained and expanding research activity in this field.

In general, two approaches are commonly used for radiomics: hand-crafted and deep learning-based methods [65]. In hand-crafted radiomics, regions of interest (ROIs) are manually segmented, after which radiomic features are extracted. Radiomics analysis typically involves five major steps [66], i.e., image segmentation, image processing, feature extraction, feature selection, and model building. Automatic segmentation methods, particularly deep learning approaches, have increasingly been applied to improve accuracy and efficiency. Here, we briefly review MRI segmentation methods using deep learning methods.

## 5. MR Image Segmentation

A widely used deep learning architecture is the U-Net framework [67], a symmetric encoder–decoder convolutional neural network (CNN) with skip connections designed for precise pixel-level segmentation, particularly in biomedical imaging. Variants such as UNet++ [68] have been developed to reduce the semantic gap between encoder and decoder feature maps, enhancing segmentation accuracy. The nnU-Net framework (“no-new U-Net”) [69] automatically configures preprocessing, network architecture, training, and postprocessing for new medical segmentation tasks, eliminating the need for manual tuning.

Other deep learning approaches include UNesT [70], which employs hierarchical transformers for efficient local spatial representation learning in 3D medical image segmentation, outperforming nnU-Net in some tasks. Encoder–decoder architectures such as V-Net and SegNet have also been widely used [71,72]. V-Net incorporates residual blocks to improve training stability, while SegNet, based on VGG-16 layers, stores max-pooling indices in the decoder instead of using skip connections, reducing memory requirements and improving efficiency.

Attention U-Net [73] enhances segmentation of small or complex structures by incorporating attention gates in skip connections, automatically weighting feature maps to suppress irrelevant regions. Recently, segmentation models such as the Segmentation Anything Model [74] and whole-body tools like TotalSegmentor [75] and MONAI Label [76] have been applied in medical imaging. For detailed surveys on deep learning–based medical image segmentation, see [77].

Designing deep learning segmentation models requires careful consideration of both network architecture and the choice of loss function. Common loss functions include Dice score, weighted Dice, and cross-entropy (CE). Focal and Focal Tversky losses are particularly effective for small lesions or tumors, while boundary, Hausdorff, and surface losses are used for contour precision. Hybrid losses (e.g., Dice + CE, Dice + Focal) often outperform single loss functions. Many other loss functions have also been proposed; see [78,79] for comprehensive surveys of loss functions in semantic segmentation.

Once ROIs have been segmented, radiomic features can be extracted from both the image and the corresponding mask. Several open-source software tools are available for feature extraction (e.g., https://theibsi.github.io/implementations, accessed on 15 December 2025), and commercial platforms such as the MATLAB Medical Imaging Toolbox (version R2022b or later versions) also provide radiomic feature extraction functions. To ensure that radiomic models can be reused in other studies, features should comply with the Image Biomarker Standardization Initiative guidelines (https://theibsi.github.io/ibsi1, accessed on 15 December 2025) [80], which standardize radiomic features and facilitate reproducibility of machine learning models across studies.

In hand-crafted radiomics, segmentation—whether manual or semi-automatic—is often tedious and time-consuming. Moreover, results may be subjective, depending heavily on expert knowledge. Deep learning approaches for radiomics study, including CNNs, have been increasingly applied for MRI-based classification studies [81]. Various network architectures have been developed for classification tasks [82,83], including autoencoder-based methods [84]. In autoencoder approaches, deep features are first extracted from images, and classification is then performed based on these features. Comprehensive reviews on this topic are available [85,86].

## 6. Machine Learning Methods and Model Selection for Radiomics Study

In hand-crafted radiomics, after MR image segmentation and feature extraction, the next critical step is to develop machine learning models. Since the number of radiomic features often exceeds the number of subjects, feature selection is essential to build effective models [87]. Regularization methods are commonly used to simplify the model while maintaining high performance, such as accuracy. For linear model/feature selection, LASSO (Least Absolute Shrinkage and Selection Operator) is frequently applied [88]. Other feature selection strategies, such as sequential model selection, involve adding or removing features or hyperparameters iteratively, evaluating model performance at each step using criteria like cross-validation, until an optimal model is achieved or performance plateaus. Compared with LASSO method for model selection, sequential methods are computationally slower. Different machine learning algorithms often require specific feature selection techniques [89].

Machine learning classification methods can be broadly categorized as unsupervised, such as k-means clustering for MRI data [90], or supervised, which rely on labeled data. Common supervised machine learning classification approaches include Naive Bayes, support vector machines, XGBoost, deep learning methods, logistic regression, TabPFN [91], and multilayer perceptron classifiers [92]. Selecting the most appropriate classifier with associate hyperparameters for a specific problem remains a challenge. To overcome this, automated model selection frameworks with optimized hyperparameters have been implemented in software such as MATLAB [93].

Beyond classification, MRI radiomics has also been applied in regression tasks. For example, radiomic features have been used for survival analysis using Cox regression models [94], progression-free survival studies, and recurrence time estimation. Deep survival models and nomogram-based approaches have been employed to predict survival times [66,95]. Radiomic features can also support competing risk models, which are useful in clinical studies to account for multiple potential causes of an event—for example, distinguishing death caused by lung cancer versus brain cancer in patients with multiple malignancies.

In addition to conventional sequences, this review will not cover the analysis of other MRI modalities. In MRI studies, neuromelanin-sensitive MRI [96,97], magnetic resonance angiography [98], and proton magnetic resonance spectroscopy (1H MRS) [99] have been applied for various research and clinical purposes. 1H MRS provides non-invasive quantification of biochemical compounds in vivo, enabling monitoring of disease progression and treatment, with applications in disorders such as Alzheimer’s disease [100].

## 7. Discussion and Conclusions

This review covers advanced MRI analysis techniques that have emerged in recent years. It includes a broad range of MRI modalities, from fMRI to qMRI, as well as methods such as deep learning-based medical image segmentation and radiomics with machine learning approaches. For each research area, this review provides a broad yet concise summary of recent developments, without delving into extensive detail, as this rapidly evolving field is driven largely by advances in statistical modeling and deep learning methods.

For example, in MR image segmentation, various deep learning architectures—including U-Net, V-Net, and SegNet—have been developed for medical image analysis, along with specialized loss functions to improve segmentation performance. More recently, transformer-based architectures, such as UNETR [101], have been proposed and demonstrated superior performance compared to nnU-Net [102].

In MRI analysis, radiomics—based on either manual or automated segmentation—has emerged as a powerful approach for quantitative imaging assessment. Both supervised and unsupervised machine learning techniques have been applied to regression and classification tasks. However, overfitting remains a common challenge, highlighting the need for careful model selection and validation. In regression studies, an important direction is the integration of clinical variables with radiomic features for survival analysis.

Class imbalance is another major issue that affects both regression and classification models [103]. In deep learning, class imbalance continues to pose open challenges, requiring further methodological innovations to ensure robust and reliable model performance [104].

Overall, the integration of advanced MRI analysis and machine learning holds great potential for driving clinical research forward and improving patient care.

## 8. Future Research Directions

One key direction is the application of deep learning to MRI analysis, moving beyond narrow, task-specific tools toward scalable, generalizable models. Advances in segmentation and classification will leverage foundation models trained on large, multi-center datasets, enabling fine-tuning across diverse imaging tasks with minimal additional data and model adjustment.

As high-resolution data from high-field MRI scanners become more widely available, there is a need for faster and more accurate qMRI calculation and analysis methods. When combined with clinical information, qMRI has the potential to generate reliable, reproducible biomarkers that bridge imaging data with clinical outcomes.

Integration of imaging with complementary data modalities, including genetic, wearable, and clinical data, as well as incorporation of deep learning with large language models, will enable richer diagnostic insights and clinically relevant narratives to support decision-making. A critical focus will be on ensuring robustness across sites, scanners, and patient populations through domain adaptation, uncertainty quantification, and rigorous external validation.

Clinical deployment of MRI analysis methods promises to improve workflow efficiency and patient care. However, achieving meaningful clinical impact remains challenging. Many current studies are retrospective, rely on dataset availability rather than clinical relevance, and report improvements smaller than the evaluation error, limiting generalizability to real-world settings [105,106].

Other challenges include the lack of balanced and well-labeled datasets, susceptibility of deep learning models to adversarial attacks, image noise, low user and patient trust, and ethical or privacy concerns [107]. Addressing data privacy, bias, and the need for expert oversight is essential for responsible and effective use of artificial intelligence in healthcare [108].

With faster scanning, higher resolution [109], automated post-processing, and multimodal integration [110], MRI analysis is expected to play an increasingly important role in clinical decision-making. For example, deep learning-based MRI analysis has been applied to brain tumor classification, segmentation, and prognosis, although challenges remain in data scarcity, generalizability, and interpretability in neuro-oncology studies [111]. Overall, the next generation of MRI acquisition, processing, and analysis aims to deliver high accuracy, speed, and robustness, but substantial work remains to realize this potential.

## Figures and Tables

**Figure 1 diagnostics-15-03212-f001:**
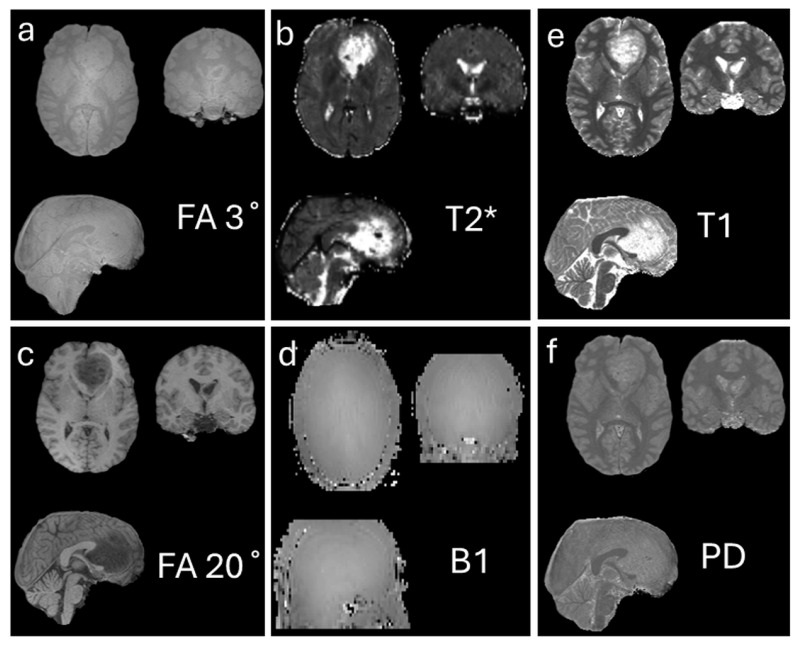
Generation of T1 (**e**) and proton density (PD) (**f**) maps using the flip angle method. (**a**,**c**) show images acquired with flip angles of 3° and 20°, respectively, which are used as input to the flip angle calculation. (**b**) T2* map estimated from a multi-echo sequence, providing additional tissue relaxation information. (**d**) B1 map estimated to correct for inhomogeneities in the transmit radiofrequency field. Together, these data allow accurate computation of the quantitative T1 and PD maps shown in (**e**,**f**). The data used in this figure were generated by the author as part of an unpublished study.

**Figure 2 diagnostics-15-03212-f002:**
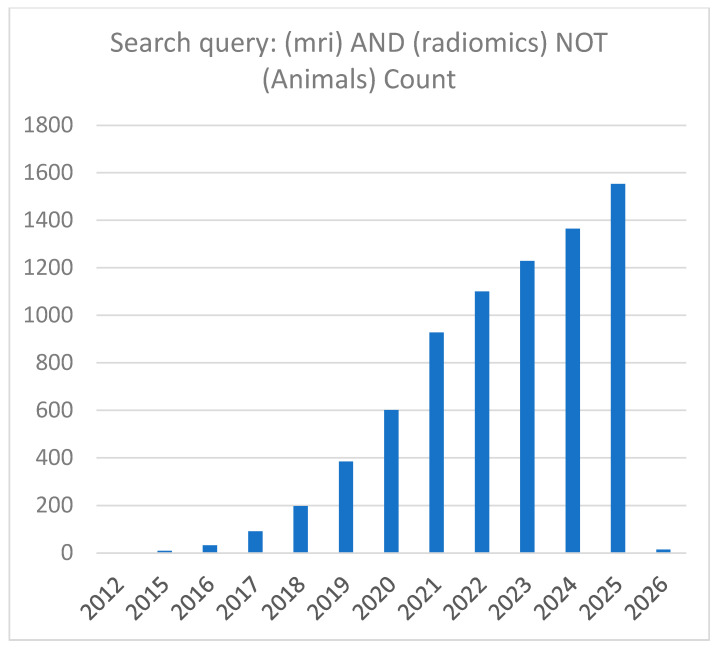
Number of published MRI radiomics papers indexed in PubMed. The x-axis represents the publication year, and the y-axis represents the number of papers. This overview shows the yearly trend of papers collected for this survey on MRI radiomics study. Data are current as of 7 December 2025.

## Data Availability

No new data were created or analyzed in this study. Data sharing is not applicable to this article.

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
