# Peer review of "Functional, Structural, and AI-Based MRI Analysis: A Comprehensive Review of Recent Advances"

_diagnostics, 2025, doi:10.3390/diagnostics15243212_

Round 1

Reviewer 1 Report

Comments and Suggestions for Authors

This is a review article on the use of artificial intelligence and MRI images in medical imaging science. In this review study, the researchers have surveyed the latest medical imaging MRI methods and artificial intelligence. In my opinion, it requires major revisions. The overall conclusion and the study's objective need to be rewritten. A list of my comments is provided below.

1-The study's objective is too broad, encompassing everything from various MRI imaging methods to artificial intelligence techniques. It is suggested that the study's objective be narrowed down to a specific topic, and the study should be rewritten accordingly.

2-The title of the study is not representative of its content and should be modified.

3-Regarding the literature search, precise statistics on the number of articles searched, rejected, and selected should be added.

4-It is recommended that in the qMRI studies section, studies on T1 and T2 mapping used for investigating glioma tumor grades be added to the study.

5-In the qMRI section, it is suggested that past studies be examined in a more targeted manner, for example, focusing on a specific disease, to provide readers with decisive insights.

6-What was the objective of this study? Have you ultimately achieved these objectives?

7-The discussion and conclusion section needs to be rewritten, as the current discussion and conclusion are very vague and general and do not follow a specific objective.

Author Response

This is a review article on the use of artificial intelligence and MRI images in medical imaging science. In this review study, the researchers have surveyed the latest medical imaging MRI methods and artificial intelligence. In my opinion, it requires major revisions. The overall conclusion and the study's objective need to be rewritten. A list of my comments is provided below.

1-The study's objective is too broad, encompassing everything from various MRI imaging methods to artificial intelligence techniques. It is suggested that the study's objective be narrowed down to a specific topic, and the study should be rewritten accordingly.

In accordance with the requirements of this journal, the review article's scope covers a broad spectrum of magnetic resonance imaging (MRI) topics. The author aimed to include diverse research methodologies within the field to both encourage further investigation and gauge interest from professionals actively involved in MRI analysis research. To accommodate these varied requirements, the author will maintain this broad topical range, allowing readers interested in a specific subject to trace relevant sources and delve deeper into the topic.

2-The title of the study is not representative of its content and should be modified.

Thank you for suggesting this. If the section editor allows, it could be replaced with the following title:

Functional, Structural, and AI-Based MRI Analysis: A Comprehensive Review of Recent Advances.

3-Regarding the literature search, precise statistics on the number of articles searched, rejected, and selected should be added.

The author accepts the reviewer's point. Nevertheless, acquiring the rejection data for papers submitted to different journals presents a significant challenge. Conversely, PubMed provides a reliable and substantial data source for analyzing the research topic's trends.

4-It is recommended that in the qMRI studies section, studies on T1 and T2 mapping used for investigating glioma tumor grades be added to the study.

A few more references have been added to the research topic.

Study showed that the T1 map has the potential to use in the clinic for glioma grading purposes due to their high performance. https://pubmed.ncbi.nlm.nih.gov/36473544/

With logistic regression, another study showed that T2 mapping also improve for glioma grading (https://www.sciencedirect.com/science/article/pii/S2352914823000436 )

5-In the qMRI section, it is suggested that past studies be examined in a more targeted manner, for example, focusing on a specific disease, to provide readers with decisive insights.

The following have been included.

As an example for brain tumour study, it showed that qMRI has potential to improve disease characterization in brain tumor patients under tight clinical time-constraints.

https://pubmed.ncbi.nlm.nih.gov/36473544/

https://www.sciencedirect.com/science/article/pii/S2352914823000436

6-What was the objective of this study? Have you ultimately achieved these objectives?

One objective of this study is to provide an overview of research in the field of MRI data analysis. The author aims to summarize recent developments and highlight potential research directions, particularly those that have emerged in recent years.

Another objective is to present a diverse range of research methodologies within the field, both to encourage further investigation and to assess the level of interest among professionals actively engaged in MRI research.

7-The discussion and conclusion section needs to be rewritten, as the current discussion and conclusion are very vague and general and do not follow a specific objective.

It has been modified as follows.

This review covers advanced MRI analysis techniques that have emerged in recent years. It includes a broad range of MRI modalities, from fMRI to qMRI, as well as methods such as deep learning–based medical image segmentation and radiomics with machine learning approaches. For each research area, the review provides a concise summary of recent developments, without delving into extensive detail, as this rapidly evolving field is driven largely by advances in statistical modeling and deep learning.

For example, in MR image segmentation, various deep learning architectures—including U-Net, V-Net, and SegNet—have been developed for medical image analysis, along with specialized loss functions to improve segmentation performance. More recently, transformer-based architectures, such as UNETR, have been proposed and demonstrated superior performance compared to nnU-Net(https://openaccess.thecvf.com/content/WACV2022/papers/Hatamizadeh_UNETR_Transformers_for_3D_Medical_Image_Segmentation_WACV_2022_paper.pdf).

In MRI analysis, radiomics—based on either manual or automated segmentation—has emerged as a powerful approach for quantitative imaging assessment. Both supervised and unsupervised machine learning techniques have been applied to regression and classification tasks. However, overfitting remains a common challenge, highlighting the need for careful model selection and validation. In regression studies, an important direction is the integration of clinical variables with radiomic features for survival analysis.

Class imbalance is another major issue that affects both regression and classification models (https://ieeexplore.ieee.org/abstract/document/10845793). In deep learning, class imbalance continues to pose open challenges, requiring further methodological innovations to ensure robust and reliable model performance (https://link.springer.com/article/10.1007/s10994-022-06268-8).

Overall, the integration of advanced MRI analysis and machine learning holds great potential for driving clinical research forward and improving patient care.

Reviewer 2 Report

Comments and Suggestions for Authors

The study's topic is interesting. However, it was studied pretty extensively. The current article is a narrative review and does not provide new information for the readers. The audience will be losing interest. Please see the other comments, 

  1. Why the weighted MRI images are missing the physical meaning? It is misleading since the data is used for computing the maps.
  2. Please mention the source of the MRI data in Fig.1.
  3. You're missing 3-point dixon as well.
  4. Medical imaging toolbox is introduced with 2022b not 2024b. Please correct it.

Author Response

Comments and Suggestions for Authors

The study's topic is interesting. However, it was studied pretty extensively. The current article is a narrative review and does not provide new information for the readers. The audience will be losing interest.

Thank you for pointing these out. I agree with the reviewer’s comments. Given the journal’s title and requirements, the review topic necessarily covers a wide range of studies within the MRI data analysis field. As a result, the authors must mention several research directions across different areas, which inevitably broadens the scope of the review.

Please see the other comments, 

  1. Why the weighted MRI images are missing the physical meaning? It is misleading since the data is used for computing the maps.

Weighted MRI (wMRI) images lack true physical meaning compared to quantitative MRI (qMRI) because wMRI reflects relative signal intensities influenced by multiple intertwined factors, rather than a single measurable property. In contrast, qMRI provides absolute numerical values corresponding to specific biophysical parameters. For radiomics studies, wMRI data must therefore be normalized before being used in machine learning classification models to enable meaningful comparison and analysis.

  1. Please mention the source of the MRI data in Fig.1.

The data used in this figure were generated by the author as part of an unpublished study.

  1. You're missing 3-point dixon as well.

The following have been added to the manuscript.

The original 2-point Dixon method used only two images (in-phase and out-of-phase), but it was sensitive to magnetic field inhomogeneity. To remedy this, the 3-point (multiple-point) method adds a third echo time, allowing the system to estimate and correct for field inhomogeneities, greatly improving robustness.

https://link.springer.com/article/10.1007/s00330-025-11564-7

https://www.mdpi.com/2813-0413/3/2/13

  1. Medical imaging toolbox is introduced with 2022b not 2024b. Please correct it.

Thanks for pointing out this. I corrected this now.

Round 2

Reviewer 1 Report

Comments and Suggestions for Authors

Dear Author

the revised manuscript improved significantly, but still some issue (comment number 1 and 3 of my previous comments).

[1-The study's objective is too broad, encompassing everything from various MRI imaging methods to artificial intelligence techniques. It is suggested that the study's objective be narrowed down to a specific topic, and the study should be rewritten accordingly.]

[3-Regarding the literature search, precise statistics on the number of articles searched, rejected, and selected should be added.]

please answer and provide update the manuscript based on these comment for more improvement.

Best regards.
